# Efficient Resource Allocation with Fairness Constraints in Restless Multi-Armed Bandits

**Dexun Li**[1]                    **Pradeep Varakantham**[1]

[1]School of Computing and Information Systems, Singapore Management University, Singapore

## Abstract

Restless Multi-Armed Bandits (RMAB) is an apt model to represent decision-making problems in public health interventions (e.g., tuberculosis, maternal, and child care), anti-poaching planning, sensor monitoring, personalized recommendations and many more. Existing research in RMAB has contributed mechanisms and theoretical results to a wide variety of settings, where the focus is on maximizing expected value. In this paper, we are interested in ensuring that RMAB decision making is also fair to different arms while maximizing expected value. In the context of public health settings, this would ensure that different people and/or communities are fairly represented while making public health intervention decisions. To achieve this goal, we formally define the fairness constraints in RMAB and provide planning and learning methods to solve RMAB in a fair manner. We demonstrate key theoretical properties of fair RMAB and experimentally demonstrate that our proposed methods handle fairness constraints without sacrificing significantly on solution quality.

## 1 INTRODUCTION

Picking the right time and manner of limited interventions is a problem of great practical importance in tuberculosis [Mate et al., 2020], maternal and child care [Biswas et al., 2021, Mate et al., 2021b], anti-poaching operations [Qian et al., 2016], cancer detection [Lee et al., 2019], and many others. All these problems are characterized by multiple arms (i.e., patients, pregnant mothers, regions of a forest) whose state evolves in an uncertain manner (e.g., medication usage in the case of tuberculosis, engagement patterns of mothers on calls related to good practices in pregnancy) and threads moving to "bad" states have to be steered to

"good" outcomes through interventions. The key challenge is that the number of interventions is limited due to a limited set of resources (e.g., public health workers, patrol officers in anti-poaching operations). Restless Multi-Armed Bandits (RMAB), a generalization of Multi-Armed Bandits (MAB) that allows non-active bandits to also undergo the Markovian state transition, has become an ideal model to represent the aforementioned problems of interest as it models uncertainty in arm transitions (to capture uncertain state evolution), actions (to represent interventions) and budget constraint (to represent limited resources).

Existing work [Mate et al., 2020, Biswas et al., 2021, Mate et al., 2021a] has focused on developing theoretical insights and practically efficient methods to solve RMAB. At each decision epoch, RMAB methods identify arms that provide the biggest improvement with an intervention. Such an approach though technically optimal can result in certain arms (or type of arms) getting starved for interventions.

In the case of interventions with regards to public health, RMAB algorithms focus interventions on the top beneficiaries who will improve the objective (public health outcomes) the most. This can result in certain beneficiaries never talking to public health workers and thereby moving to bad states (and potentially also impacting other beneficiaries in the same community) from where improvements can be minor even with intervention and hence never getting picked by RMAB algorithms. As shown in Fig. 1, when using the Threshold Whittle index approach proposed by Mate et al. [2020], the arm activation probability is lopsided, with 30% of arms getting activated more than 50 times and 50% of the arms are never activated. Such starvation of interventions can result in arms moving to a bad state from where interventions cannot provide big improvements and therefore there is further starvation of interventions for those arms. Such starvation can happen to entire regions or communities, resulting in lack of fair support for beneficiaries in those regions/communities. To avoid such cycles between bad outcomes, there is a need for RMAB algorithms to consider fairness in addition to maximizing expected reward when

*Accepted for the 38th Conference on Uncertainty in Artificial Intelligence* (UAI 2022).

picking arms. Risk sensitive RMAB [Mate et al., 2021b] considers an objective that targets to reduce such starvation, however, they *do not guarantee* that arms (or types of arms) are picked a minimum number of times.

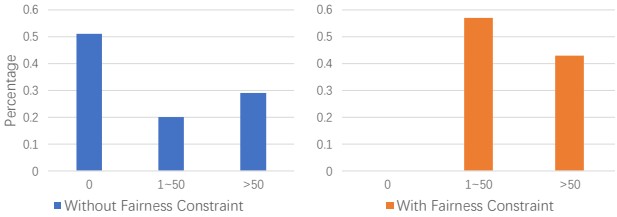

Figure 1: The x-axis is the number of times activated, and the y-axis is the percentage of each frequency range. We consider the RMAB given in Section 2, with $k = 10$, $N = 100$, $T = 1000$ and $L = 50$, $\eta = 2$. Left: the result of using the Whittle index algorithm without considering fairness constraints. Right: the result of when considering fairness constraints. As can be noted, without fairness constraints in place, almost 50% of the arms never get activated.

Recent work in Multi-Armed Bandits (MAB) has presented different notions of fairness. For example, Li et al. [2019] study a Combinatorial Sleeping MAB model with Fairness constraints, called CSMAB-F. The fairness constraints ensure a minimum selection fraction for each arm. Patil et al. [2020] introduce similar fairness constraints in the stochastic multi-armed bandit problem, where they use a pre-specified vector to denote the guaranteed number of pulls. Joseph et al. [2016] define fairness as saying that a worse arm should not be picked compared to a better arm, despite the uncertainty on payoffs. Chen et al. [2020] define the fairness constraint as a minimum rate that is required when allocating a task or resource to a user. The above fairness definitions are relevant and we generalize from these to propose a fairness notion for RMAB. Unfortunately, approaches developed for fair MAB cannot be utilized for RMAB, due to uncertain state transitions with passive actions as well.

**Contributions:** To the best of our knowledge, this is the first paper to consider fairness constraints in RMAB. Here are the key contributions:

- We propose a fairness constraint wherein for any arm (or more generally, for a type of arm), we require that the number of decision epochs since the arm (or the type of arm) was activated last time is upper bounded. This will ensure that every arm (or type of arm) gets activated a minimum number of times, thus generalizing on the fairness notions in MAB described earlier.

- We provide a modification to the Whittle index algorithm that is scalable and optimal while being able to handle both finite and infinite horizon cases. We also provide a model-free learning method to solve the problem when the transition probabilities are not known

beforehand.

- Experiment results on the generated dataset show that our proposed approaches can achieve good performance while still satisfying the fairness constraint.

## 2 PROBLEM DESCRIPTION

In this section, we formally introduce the RMAB problem. There are $N$ independent arms, each of which evolves according to an associated Markov Decision Process (MDP). An MDP is characterized by a tuple $\{\mathcal{S}, \mathcal{A}, \mathcal{P}, r\}$, where $\mathcal{S}$ represents the state space, $\mathcal{A}$ represents the action space, $\mathcal{P}$ represents the transition function, and $r$ is the state-dependent reward function. Specifically, each arm has a binary-state space: 1 ("good") and 0 ("bad"), with action-dependent transition matrix $\mathcal{P}$ that is potentially different for each arm. Let $a_t^i \in \{0, 1\}$ denote the action taken at time step $t$ for arm $i$, and $a_t^i = 1(a_t^i = 0)$ indicates an active (passive) action for arm $i$. Due to limited resources, at each decision epoch, the decision-maker can activate (or intervene on) at most $k$ out of $N$ arms and receive reward accrued from all arms determined by their states. $\sum_{i=1}^{N} a_t^i = k$ describes this limited resource constraint. Figure 2 provides an example of an arm in RMAB.

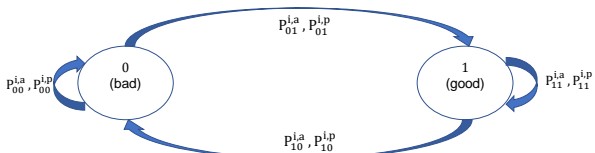

Figure 2: $a$ and $p$ denote the active and passive actions on arm $i$ respectively. $P_{s,s'}^{i,a}$ and $P_{s,s'}^{i,p}$ are the transition probabilities from state $s$ to state $s'$ under action $a$ and $p$ respectively for arm $i$.

The state of arm $i$ evolves according to the transition matrix $P_{s,s'}^{a,i}$ for the active action and $P_{s,s'}^{p,i}$ for the passive action. We follow the setting in Mate et al. [2020], when the arm $i$ is activated, the latent state of arm $i$ will be fully observed by the decision-maker. The states of passive arms are unobserved by the decision-maker.

When considering such partially observable problem, it is sufficient to let the MDP state be the belief state: the probability that the arm is in the "good" state. We need to keep track of the belief state on the current state of the unobserved arm. This can be derived from the decision-maker's partial information which is encompassed by the last observed state and the number of decision time steps since the last activation of the arm. Let $\omega_s^i(u)$ denote the belief state, i.e., the probability that the state of arm $i$ is 1 when it was activated $u$ time steps ago with the observed state $s$. The belief state in next time step can be obtained by solving the

following recursive equations:

$$\omega_s^i(u+1) = \begin{cases} \omega_s^i(u)P_{1,1}^{p,i} + (1-\omega_s^i(u))P_{0,1}^{p,i} & \text{passive} \\ P_{s',1}^{i,a} & \text{active} \end{cases}$$
(1)

Where $s'$ is the new state observed for arm $i$ when the active action was taken. The belief state can be calculated in closed form with the given transition probabilities. We let $\omega = \omega_s^i(u+1)$ for ease of explanation when there is no ambiguity.

A policy $\pi$ maps the belief state vector $\Omega_t = \{\omega_t^1, \cdots, \omega_t^N\}$ at each time step $t$ for all arms to the action vector, $a_t = \{0,1\}^N$. Here $\omega_t^i$ is the belief state for arm $i$ at time step $t$. We want to design an optimal policy to maximize the cumulative long-term reward over all the arms. One widely used performance measure is the expected discounted reward over the horizon $T$:

$$\mathbb{E}_\pi[\sum_{t=1}^T \beta^{t-1} R_t(\Omega_t, \pi(\Omega_t))|\Omega_0]$$

Here $R_t(\Omega_t, \pi(\Omega_t))$ is the reward obtained in slot $t$ under action $a_t = \pi(\Omega_t)$ determined by policy $\pi$, $\beta$ is the discount factor. As we discussed in the introduction, in addition to maximizing the cumulative reward, ensuring fairness among the arms is also a key design concern for many real-world applications. In order to model the fairness requirement, we introduce constraints that ensure that any arm (or kind of arms) is activated at least $\eta$ times during any decision interval of length $L$. The overall optimization problem corresponding to the problem at hand is thus given by:

$$\text{maximize}_\pi \ \mathbb{E}_\pi[\sum_{t=1}^T \beta^{t-1} R_t(\Omega_t, \pi(\Omega_t))|\Omega_0]$$
$$\text{subject to } \sum_i^N a_t^i = k, \forall t \in \{1,\ldots,T\}$$
$$\sum_{t=u}^{u+L} a_t^i \geq \eta \quad \forall u \in \{1,\ldots,T-L\}, \forall i \in \{1,\ldots,N\}.$$
(2)

$\eta$ is the minimum number of times an arm should be activated in a decision period of length $L$. The strength of fairness constraints is thus governed by the combination of $L$ and $\eta$. Obviously, this requires $k \times L > N \times (\eta - 1)$ as the fairness constraint should meet the resource constraint. This fairness problem can be formulated at the level of regions/communities by also summing over all the arms, $i$ in a region in the second constraint, i.e.,

$$\sum_{i \in r} \sum_{t=u}^{u+L} a_t^i \geq \eta$$

Our approaches with a simple modification are also applicable to this fairness constraint at the level of regions/communities.

## 3 BACKGROUND: WHITTLE INDEX

In this section, we describe the Whittle Index algorithm [Whittle, 1988] to solve RMAB. This algorithm at every time step, computes index values (Whittle Index values) for every arm and then activates the arms that have the top "$k$" index values. Whittle index quantifies how appealing it is to activate a certain arm. This algorithm provides optimal solutions if the underlying RMAB satisfies the indexability property, defined in Definition 1.

Formally[1], the Whittle index of an arm in a belief state $\omega$ (i.e., the probability of good state 1) is the minimum subsidy $\lambda$ such that it is optimal to make the arm passive in that belief state. Let $V_{\lambda,T}(\omega)$ denote the value function for the belief state $\omega$ over a horizon $T$. Then it could be written as

$$V_{\lambda,T}(\omega) = \max\{V_{\lambda,T}(\omega; a=0), V_{\lambda,T}(\omega; a=1)\}, \quad (3)$$

where $V_{\lambda,T}(\omega; a=0)$ and $V_{\lambda,T}(\omega; a=1)$ denote the value function when taking passive and active actions respectively at the first decision epoch followed by optimal policy in the future time steps. Because the expected immediate reward is $\omega$ and subsidy for a passive action is $\lambda$, we have the value function for passive action as:

$$V_{\lambda,T}(\omega, a=0) = \lambda + \omega + \beta V_{\lambda,T-1}(\tau^1(\omega)), \quad (4)$$

where $\tau^1(\omega)$ is the 1-step belief state update of $\omega$ when the passive arm is unobserved for another 1 consecutive slot (see the update rule in Eq. 1). Note that $\omega$ is also the expected reward associated with that belief state. For an active action, the immediate reward is $\omega$ and there is no subsidy. However, the actual state will be known and then evolve according to the transition matrix for the next step:

$$V_{\lambda,T}(\omega, a=1) = \omega + \beta(\omega V_{\lambda,T-1}(P_{1,1}^a) + (1-\omega)V_{\lambda,T-1}(P_{0,1}^a)). \quad (5)$$

**Definition 1** *An arm is indexable if the passive set under the subsidy $\lambda$ given as $\mathcal{P}_\lambda = \{\omega : V_{\lambda,T}(\omega, a = 0) \geq V_{\lambda,T}(\omega, a = 1)\}$ monotonically increases from $\emptyset$ to the entire state space as $\lambda$ increases from $-\infty$ to $\infty$. The RMAB is indexable if every arm is indexable.*

Intuitively, this means that if an arm takes passive action with subsidy $\lambda$, it will also take passive action if $\lambda' > \lambda$. Given the *indexability*, $W_T(\omega)$ is the least subsidy, $\lambda$ that makes it equally desirable to take active and passive actions.

$$W_T(\omega) = \inf_\lambda \{\lambda : V_{\lambda,T}(\omega; a=1) \leq V_{\lambda,T}(\omega; a=0)\}$$
(6)

---

[1]Since we will only be talking about one arm at a time step, we will abuse the notation by not indexing belief, action and value function with arm id or time index.

**Definition 2** *A policy is a threshold policy if there exists a threshold $\lambda_{th}$ such that the action is passive $a = 0$ if $\lambda > \lambda_{th}$ and $a = 1$ otherwise.*

Existing efficient methods for solving RMABs derive these threshold policies.

## 4 FAIRNESS IN RMAB

The key advantage of a Whittle index based approach is scalability without sacrificing solution quality. In this section, we provide Whittle index based approaches to handle fairness constraints under known and unknown transition models, with both infinite and finite horizon settings. We specifically consider partially observable settings[2].

### 4.1 INFINITE HORIZON

When we need to consider the partial observability of the state of the RMAB problem, it is sufficient to let the MDP state be the belief state: the probability that the arm is in the "good" state [Kaelbling et al., 1998]. As a result, the partially observable RMAB has a large number of belief states [Mate et al., 2020].

Recall that the definition of the Whittle index $W_T(\omega)$ of belief state $\omega$ is the smallest $\lambda$ s.t. it is optimal to make the arm passive in the current state. We can compute the Whittle index value for each arm, and then rank the index value of all $N$ arms and select top $k$ arms at each time step to activate. With fairness constraints, the change to the approach is minimal and intuitive. ***The optimal policy is to choose the arms with the top "k" index values until a fairness constraint is violated for an arm. In that time step, we replace the last arm in top-$k$ with the arm for which fairness constraint is violated.*** We show that this simple change works across the board for the infinite and finite horizon, fully and partially observable settings. We provide the detailed algorithm in Algorithm 1 and also provide sufficient conditions under which the Algorithm 1 is optimal.

We now provide the expression for $\lambda$. $V_{\lambda,\infty}(\omega)$ denotes the value that can be accrued from a single-armed bandit process with subsidy $\lambda$ over infinite time horizon ($T \to \infty$) if the belief state is $\omega$. Therefore, we have:

$$V_{\lambda,\infty}(\omega) = \max \begin{cases} \lambda + \omega + \beta V_{\lambda,\infty}(\tau^1(\omega)) & \text{passive} \\ \omega + \beta \left( \omega V_{\lambda,\infty}(P^a_{1,1}) + (1-\omega)V_{\lambda,\infty}(P^a_{0,1}) \right) & \text{active} \end{cases} \tag{7}$$

For any belief state $\omega$, the $u$-steps belief update $\tau^u(\omega)$ will converge to $\omega^*$ as $u \to \infty$, where $\omega^* = \frac{P^p_{0,1}}{1 + P^p_{0,1} - P^p_{1,1}}$. It should be noted that this convergence can happen in two ways depending on the state transition patterns:

---
[2]We also provide a discussion about fully observable setting in the appendix

- Case 1: Positively correlated channel $(P^p_{1,1} \geq P^p_{0,1})$. The belief update process is shown in Figure 3. We can see that for the positively correlated case, they have a monotonous belief update process.

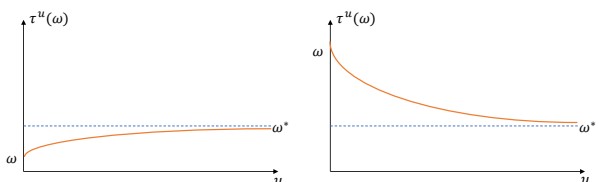

Figure 3: The $u$-step belief update of an unobserved arm $(P^p_{1,1} \geq P^p_{0,1})$

We first consider the *non-increasing belief process* as indicated in the right graph. Formally, for $\forall u \in \mathbb{N}^+$, we have $\omega(u) \geq \omega(u + 1)$ if the initial belief state $\omega$ is above the convergence value. Similarly, for the *increasing belief process* shown in the left graph, we have the initial belief state $\omega < \omega^*$.

- Case 2: Negatively correlated channel $(P^p_{1,1} < P^p_{0,1})$.

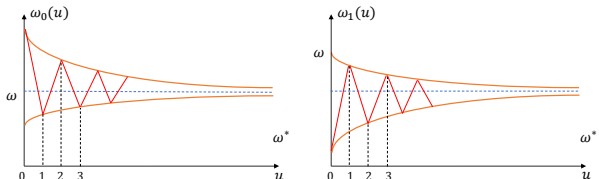

Figure 4: The $u$-step belief update of an unobserved arm $(P^p_{1,1} < P^p_{0,1})$

The belief state converges to $\omega^*$ from the opposite direction as shown in Figure 4. This case has similar properties and is less common in the real world because it is more likely to remain in a good state than to move from a bad state to a good state. Therefore, we omit the lengthy discussion.

The belief state transition patterns are of particular importance because in proving optimality of Algorithm 1, the belief evolution pattern for the arm (whose fairness constraint will be violated) plays a crucial role.

**Theorem 1** *For infinite time horizon ($T \to \infty$) RMAB with Fairness Constraints governed by parameters $\eta$ and $L$, Algorithm 1 ( i.e., activating arm $i$ at the end of the time period when its fairness constraint is violated) is optimal:*

*1. For $\omega^i \leq \omega^*$ (increasing belief process), if*

$$(P^{i,p}_{1,1} - P^{i,p}_{0,1})\left(1 + \frac{\beta\Delta_3}{1-\beta}\right)\left(1 - \beta(P^{i,a}_{1,1} - P^{i,a}_{0,1})\right)$$
$$\leq (P^{i,a}_{1,1} - P^{i,a}_{0,1}) \tag{8}$$

$\Delta_3 = \min\{(P^{i,p}_{1,1} - P^{i,p}_{0,1}), (P^{i,a}_{1,1} - P^{i,a}_{0,1})\}$.

**Algorithm 1:** Fair Whittle Thresholding (FaWT)

**Input:** Transition matrix $\mathcal{P}$, fairness constraint, $\eta$ and $L$, set of belief states $\{\omega^1, \ldots, \omega^N\}$, $k$

1 **for** *each arm $i$ in $1$ to $N$* **do**
2  Compute the corresponding Whittle index $TW(\omega^i)$ under the infinite horizon using the *Forward and Reverse Threshold* policy;
3  **if** *the activation frequency $\eta$ for arm $i$ will not be satisfied at the end of the period of length $L$* **then**
4   Add arm $i$ to the action set $\phi$;
5   $k = k - 1$;
6  **if** *finite horizon* **then**
7   Compute the the index value $W_1(\omega^i)$;
8   Compute the Whittle index $W_T(\omega^i)$ using Equation 10;
9 Add arms with top k highest $TW(\cdot)$ (for infinite horizon case) or $W_T(\cdot)$ (for finite horizon case) values to the action set $\phi$ Decrease the residual time horizon by $T = T - 1$;

**Output:** Action set $\phi$

2. *For $\omega^i \geq \omega^*$ (non-increasing belief process), if:*

$$(P_{1,1}^{i,p} - P_{0,1}^{i,p})(1 - \beta)\Delta_1 \geq$$
$$(P_{1,1}^{i,a} - P_{0,1}^{i,a})\left(1 - \beta(P_{1,1}^{i,a} - P_{0,1}^{i,a})\right) \quad (9)$$

$$\Delta_1 = \min\{1, 1 + \beta(P_{1,1}^{i,p} - P_{0,1}^{i,p}) - \beta(P_{1,1}^{i,a} - P_{0,1}^{i,a})\}$$

*Proof Sketch.* Consider an arm $i$ that has not been activated for $L - 1$ time slots. In such a case, Algorithm 1 will select arm $i$ to activate in the next time step $t = L$. Define the intervention effect of activating arm $i$ as

$$V_{\lambda,\infty}(\omega, a = 1) - V_{\lambda,\infty}(\omega, a = 0)$$

Following standard practice and for notational convenience, we do not index the intervention effect and value functions with $i$. Due to independent evolution of arms, moving active action of arm $i$ does not result in a greater value function for other arms according to the Whittle index algorithm, thus it suffices to only consider arm $i$. Here is the proof flow:
(1) Algorithm 1 optimality requires that the intervention effect at time step $t = L - 1$ is smaller than intervention effect at $t = L$. Optimality can be established by requiring the partial derivative of the intervention effect w.r.t. time step $t$ is greater than 0.
(2) However, computing this partial derivative $\frac{\partial(V_{\lambda,\infty}(\omega,a=1)-V_{\lambda,\infty}(\omega,a=0))}{\partial t}$ is difficult because value function expression is complex. We use chain rule to get:

$$\frac{\partial(V_{\lambda,\infty}(\omega, a = 1) - V_{\lambda,\infty}(\omega, a = 0))}{\partial \omega} \cdot \frac{\partial(\omega)}{\partial(t)}$$

(3) The sign of second term, $\frac{\partial \omega}{\partial t}$ is based on the belief state transition pattern described before this theorem. We then need to consider the sign of the first term,

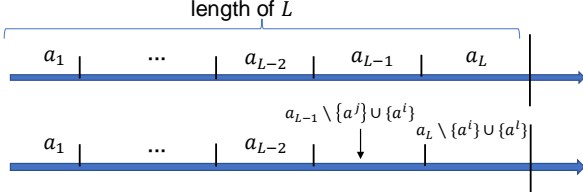

Figure 5: The action vector for RMAB is $a_t$ at time step $t$. Then we move the action $a^i$ that satisfies fairness constraint to earlier slot and replace $k$-th ranked action $a^j$. Action $a^l$ is then added according to the index value at the end.

$\frac{\partial(V_{\lambda,\infty}(\omega,a=1)-V_{\lambda,\infty}(\omega,a=0))}{\partial \omega}$.
(4) We can compute this by deriving the bound on $V_{\lambda,\infty}(\omega_1) - V_{\lambda,\infty}(\omega_2), \forall \omega_1, \omega_2$ as well as bounds on $\frac{\partial V_{\lambda,\infty}(\omega)}{\partial \omega}$. Detailed proof in appendix. $\square$

## 4.2 FINITE HORIZON

In this part, we demonstrate that the mechanism developed for handling fairness in the infinite horizon setting can also be applied to the finite horizon setting. In showing this, we address two key challenges:

1. Computing the Whittle index under the finite horizon setting in a scalable manner.

2. Showing that Whittle index value reduces as residual horizon decreases. This will assist in showing that it is optimal to activate the fairness violating arm at the absolute last step where a violation will happen and not earlier;

It is costly to compute the index under the finite horizon setting – $O(|\mathcal{S}|^k T)$ time and space complexity [Hu and Frazier, 2017]. Therefore, we take advantage of the fact that the index value has an upper and lower bound, and it will converge to the upper bound as the time horizon $T \to \infty$. Specifically, we use an appropriate functional form to approximate the index value. To do this, we first show gradual Index decay ($\lambda_T > \lambda_{T-1} > \lambda_0$) by improving on the Index decay ($\lambda_T > \lambda_0$) introduced in [Mate et al., 2021a].

**Theorem 2** *For a finite horizon $T$, the Whittle index $\lambda_T$ is the value that satisfies the equation $V_{\lambda_T,T}(\omega, a = 0) = V_{\lambda_T,T}(\omega, a = 1)$ for the belief state $\omega$. Assuming indexability holds, the Whittle index will decay as the value of horizon $T$ decreases: $\forall T > 1 : \lambda_{T+1} > \lambda_T > \lambda_0 = 0$.*

*Proof Sketch.* We can calculate $\lambda_0$ and $\lambda_1$ by solving equation $V_{\lambda_0,0}(\omega, a = 0) = V_{\lambda_0,0}(\omega, a = 1)$ and $V_{\lambda_1,1}(\omega, a = 0) = V_{\lambda_1,1}(\omega, a = 1)$ according to Eq. 4 and Eq. 5. We can then derive $\lambda_t > \lambda_{t-1}$ by obtaining $\frac{\partial \lambda_t}{\partial t} > 0$ for $\forall t > 1$ through induction method. The detailed proof can be found in the appendix. $\square$

We can easily compute $\lambda_0$, $\lambda_1$, and we have $\forall T > 1$: $\lambda_{T+1} > \lambda_T > \lambda_0 = 0$ according to Theorem 2, and $\lim_{T \to \infty} \lambda_T \to TW(\omega)$, where $TW(\omega)$ is the Whittle index value for state $\omega$ under infinite horizon. Hence, we can use a sigmoid curve to approximate the index value. One common example of a sigmoid function is the logistic function. This form is also used by Mate et al. [2021a]. Specifically, we let

$$W_T(\omega) = \frac{A}{1 + e^{-kT}} + C, \qquad (10)$$

where $A$ and $\frac{A}{2} + C$ are the curve's bounds; $k$ is the logistic growth rate or steepness of the curve. Recall that the definition of the Whittle index $W_T(\omega)$ of belief state $\omega$ is the smallest $\lambda$ s.t. it is optimal to make the arm passive in the current state. We have $W_0(\omega) = 0$ and $W_1(\omega) = \beta(\omega(P_{1,1}^a - P_{1,1}^p) + (1 - \omega)(P_{0,1}^a - P_{0,1}^p))$, and $W_\infty(\omega) = TW(\omega)$. By solving these three constraints, we can get the three unknown parameters,

$$C = -TW(\omega), A = 2TW(\omega),$$

$$k = -\log\left(\frac{2TW(\omega)}{\beta(\omega(P_{1,1}^a - P_{1,1}^p) + (1-\omega)(P_{0,1}^a - P_{0,1}^p)) + TW(\omega)} - 1\right)$$

Algorithm 1 shows how to use $W_T(\omega)$ in considering fairness constraint under the finite horizon setting. Next, we show that like in the infinite horizon case, value function and Whittle index decay over time in the case of the finite horizon.

**Theorem 3** *Consider the finite horizon RMAB problem with fairness constraint. Algorithm 1 (activating arm $i$ at the end of the time period when its fairness constraint is violated) is optimal:*

1. *When $\omega^i \leq \omega^*$ (increasing belief process), if*

$$(P_{1,1}^{i,p} - P_{0,1}^{i,p})\left(\Delta_4 \beta \sum_{t=0}^{T-2}[\beta^t] + 1\right) \leq$$

$$(P_{1,1}^{i,a} - P_{0,1}^{i,a})\sum_{t=0}^{T-2}[\beta^t(P_{1,1}^{i,a} - P_{0,1}^{i,a})^t]$$

$$(11)$$

$\Delta_4 = \min\{(P_{1,1}^{i,p} - P_{0,1}^{i,p}), (P_{1,1}^{i,a} - P_{0,1}^{i,a})\}$, *and $T$ is the residual horizon length.*

2. *When $\omega^i \geq \omega^*$ (non-increasing belief process), if*

$$(P_{1,1}^{i,p} - P_{0,1}^{i,p})\left(\Delta_2 \beta \sum_{t=0}^{T-2}[\beta^t(P_{1,1}^{i,a} - P_{0,1}^{i,a})^t] + 1\right) \geq$$

$$(P_{1,1}^{i,a} - P_{0,1}^{i,a})\sum_{t=0}^{T-2}\beta^t \qquad (12)$$

$\Delta_2 = \min\{(P_{1,1}^{i,p} - P_{0,1}^{i,p}), (P_{1,1}^{i,a} - P_{0,1}^{i,a})\}.$

*Proof Sketch.* The proof is similar to the infinite horizon case (detailed in Appendix). □

## 4.3 UNCERTAINTY IN TRANSITION MATRIX

In most real-world applications [Biswas et al., 2021], there may not be adequate information about all the state transitions. In such cases, we don't know how likely a transition is and thus, we won't be able to use the Whittle index approach directly. We provide a mechanism to apply the Thompson sampling based learning mechanism for solving RMAB problems without prior knowledge and where it is feasible to get learning experiences. Thompson sampling [Thompson, 1933] is an algorithm for online decision problems, and can be applied in MDP [Gopalan and Mannor, 2015] as well as Partially Observable MDP [Meshram et al., 2016]. In Thompson sampling, we initially assume that arm has a prior Beta distribution in the transition probability according to the prior knowledge (if there is no prior knowledge available, we assume a prior $Beta(1,1)$ as this is the uniform distribution on $(0,1)$). We choose Beta distribution because it is a convenient and useful prior option for Bernoulli rewards [Agrawal and Goyal, 2012].

In our algorithm, referred to as FaWT-U and provided in 2, at each time step, we sample the posterior distribution over the parameters, and then use the Whittle index algorithm to select the arm with the highest index value to play if the fairness constraint is not violated. We can utilize our observations to update our posterior distribution, because playing the selected arms will reveal their current state. Then, the algorithm takes samples from the posterior distribution and repeats the procedure again.

---

**Algorithm 2:** Fair Whittle Thresholding with Uncertainty in transition matrix(FaWT-U)

---
**Input:** Posterior $Beta$ distribution over the transition matrix $\mathcal{P}$, fairness constraint, $\eta$ and $L$, set of belief states $\{\omega^1, \ldots, \omega^N\}$, budget $k$

1 **for** *each arm $i$ in 1 to N* **do**
2     Sample the transition probability parameters independently from posterior;
3     Compute Whittle indices based on the transition matrix and belief state;
4 **if** *the activation frequency $\eta$ for arm $i$ is not satisfied at the end of the period of length $L$* **then**
5     Add arm $i$ to the action set $\phi$;
6     $k = k - 1$;
7 Add the arms with top $k$ index value into $\phi$;
8 Play the selected arms and receive the observations;
9 Update the posterior distribution;
**Output:** Action set $\phi$ and updated posterior distribution over parameters

---

We employ the sampled transition probabilities and belief states $\{\omega^1, \ldots, \omega^N\}$, as well as the residual time horizon $T$ as the input to the Whittle index computation (Line 3 in Algorithm 2).

## 4.4 UNKNOWN TRANSITION MATRIX

We now tackle the second challenge mentioned, in which the transition matrix is completely unknown. In this case, we can take advantage of the model-free learning method to avoid directly using the whittling index policy.

Q-Learning is most commonly used to solve the sequential decision-making problem, which was first introduced by Watkins and Dayan [1992] as an early breakthrough in reinforcement learning. It is widely studied for social good [Nahum-Shani et al., 2012, Li et al., 2021], and it has also been extensively used in RMAB problems [Fu et al., 2019, Avrachenkov and Borkar, 2020, Biswas et al., 2021] to estimate the expected Q-value, $Q^*(s, a, l)$, of taking action $a \in \{0, 1\}$ after $l \in \{1, \ldots, L\}$ time slots since last observation $s \in \{0, 1\}$. The off-policy TD control algorithm is defined as

$$Q^{t+1}(s_t, a_t, l_t) \leftarrow Q^t(s_t, a_t, l_t) +$$
$$\alpha_t(s_t, a_t, l_t) \left[ R_{t+1} + \gamma \max_a \left( Q^t(s_{t+1}, a, l_{t+1}) - Q^t(s_t, a_t, l_t) \right) \right] \quad (13)$$

Where $\gamma$ is the discount rate, $\alpha_t(s_t, a_t, l_t) \in [0, 1]$ is the learning rate parameter, i.e., a small $\alpha_t(s_t, a_t, l_t)$ will result in a slow learning process and no update when $\alpha_t(s_t, a_t, l_t) = 0$. While a large $\alpha_t(s_t, a_t, l_t)$ may cause the estimated Q-value to rely heavily on the most recent return, when $\alpha_t(s_t, a_t, l_t) = 1$, the Q-value will always be the most recent return.

We now describe how to use the Whittle index-based Q-Learning mechanism to solve the RMAB problem with fairness constraints. We build on the work by Biswas et al. [2021] for fully observable settings. In addition to considering fairness constraints, our model can be viewed as an extension to the partially observable setting. Due to fairness constraints, $l$ can be a maximum of $L$ time steps. Therefore, belief space is also limited. We are able to use the Q-Learning based approach to effectively compute the Whittle index value and this approach is summarized in Algorithm 3,

One typical form of $\alpha_t(s_t, a_t, l_t)$ could be $1/z(s_t, a_t, l_t)$, where $z(s_t, a_t, l_t) = \left( \sum_{u=0}^t \mathbb{I}\{s_u = s, a_u = a, l_u = l\} \right) + 1$ for each initial observed state $s \in \{0, 1\}$, action $a \in \{0, 1\}$ and time length since last activation $l \in \{1, \ldots, L\}$ at the time slot $u$ from the beginning. With such mild form of $\alpha_t(s_t, a_t, l_t)$, we now are able to build the theoretical support for the Q-Learning based Whittle index approach.

**Theorem 4** *Selecting the highest-ranking arms according to the $Q_i^*(s, a = 1, l) - Q_i^*(s, a = 0, l)$ till the budget constraint is met is equivalent to maximizing $\left\{ \sum_{i=1}^N Q_i^*(s, a, l) \right\}$ over all possible action set $\{0, 1\}^N$ such that $\sum_{i=1}^N a_i = k$.*

*Proof Sketch.* A proof based on work by [Biswas et al., 2021] is given in Appendix. □

---

**Algorithm 3:** Fair Whittle Thresholding based Q-Learning(FaWT-Q)

---
**Input:** parameter $\epsilon$ and $k$, and $\alpha_t(s_t, a_t, l_t)$, initial observed state set $\{s\}^N$,

1 **for** *each arm $i$ in $1$ to $N$* **do**
2    Initialize the $Q_i(s, a, l) \leftarrow 0$ for each state $s \in \{0, 1\}$, and each action $a \in \{0, 1\}$ and time length $l \in \{1, \ldots, L\}$;
3    For each $s \in \{0, 1\}$ and $l \in \{1, \ldots, L\}$ initialize the Whittle index value set $\lambda_i(s, l) \leftarrow 0$;
4 **for** *$t$ from $1$ to $T$* **do**
5    **for** *arm $i$ in $1$ to $N$* **do**
6      **if** *the fairness constraint is violated* **then**
7        Add arm $i$ to the action set $\phi$;
8        $k = k - 1$;
9    With prob $\epsilon$ add random $k$ arms to $\phi$ and with prob $1 - \epsilon$ add arms with top $k$ $\lambda_i(s, l)$ value ;
10    Activate the selected arms and receive rewards and observations;
11    **for** *each arm $i$ in $1$ to $N$* **do**
12      Update the $Q_i^{t+1}(s, a, l)$ according to Eq. 13;
13      **if** $i \in \phi$ **then**
14        Set $l = 1$ and update $s_i$ according to the received observation;
15      **else**
16        Set $l = l + 1$;
17      Update the new Q-Learning based Whittle index by $\lambda_i^{t+1}(s, l) = Q_i(s, a = 1, l) - Q_i(s, a = 0, l)$
**Output:** Action set $\phi$

---

**Theorem 5** *Stability and convergence: The proposed approach converges to the optimal with probability $1$ under the following conditions:*
*1. The state space and action space are finite;*
*2. $\sum_{t=1}^\infty \alpha_t(s_t, a_t, l_t) = \infty \quad \sum_{t=1}^\infty \alpha_t^2(\omega_i(t)) < \infty$*

*Proof Sketch.* The key to the convergence is contingent on a particular sequence of episodes observed in the real process [Watkins and Dayan, 1992]. Detailed proof is given in Appendix. □

## 5 EXPERIMENT

To the best of our knowledge, we are the first to explore fairness constraints in RMAB, hence the goal of the experiment section is to evaluate the performance of our approach in comparison to existing baselines:

**Random**: At each round, decision-maker randomly select $k$ arms to activate.

**Myopic**: Select $k$ arms that maximize the expected reward at the immediate next round. A myopic policy ignores the impact of present actions on future rewards and instead fo-

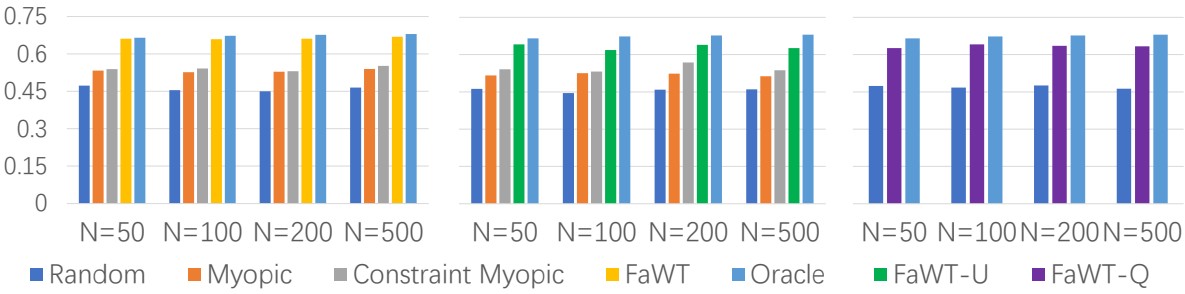

Figure 6: Comparison of performance of our approach and baseline approaches

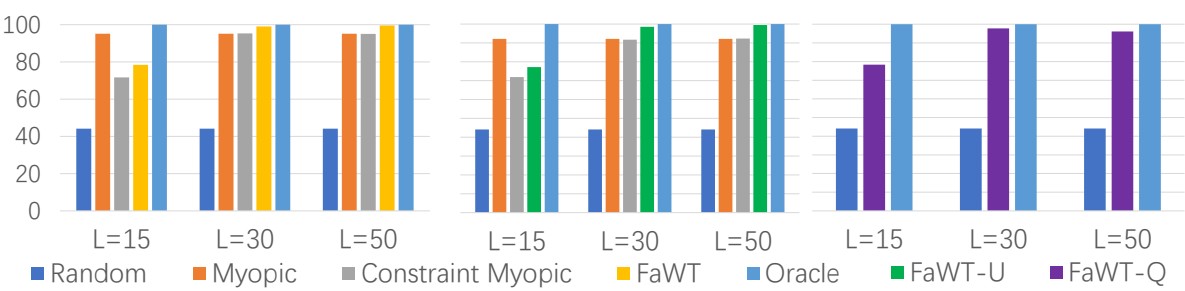

Figure 7: Intervention benefit ratio of our approach and baseline approaches without penalty for the violation of the fairness constraint. We set $N = 100$, $k = 10$, $T = 1000$, $\eta = 2$ and $L = \{15, 30, 50\}$.

cuses entirely on predicted immediate returns. Formally, this could be described as choosing the $k$ arms with the largest gap $\Delta\omega_t = (\omega_{t+1}|a_t = 1) - (\omega_{t+1}|a_t = 0)$ at time $t$.

**Constraint Myopic:** It is the same as the Myopic when there is no conflict with fairness constraints, but if the fairness constraint is violated, it will choose the arm that satisfies the fairness constraint to play.

**Oracle**: Algorithm by Qian et al. [2016] under the assumption that the states of all arms are fully observable and the transition probabilities are known without considering fairness constraints.

To demonstrate the performance of our proposed methods, we test our algorithms on synthetic domains [Mate et al., 2020] and provide numerical results averaged over 50 runs.

**Average reward value with penalty:** In Figure 6, we show the average reward $\bar{R}$ at each time step received by an arm over the time interval $T = 1000$ for $N = 50, 100, 200, 500$ and $k = 10\% \times N$ with the fairness constraint $L = 20$, and $\eta = 2$. We will receive a reward of 1 if the state of an arm is $s = 1$, and no reward otherwise. We impose a small penalty of $-0.01$ if the fairness constraint of an arm is not satisfied. The graph on the left shows the performance of FaWT method when assuming the transition matrix is known. The middle graph is the average reward obtained using the FaWT-U approach when the transition model is not fully available. The right graph illustrates the result of FaWT-Q method when the transition model is unknown. As shown in the figure, our approaches consistently outperform the Random and Myopic baselines,

and in addition to satisfying the fairness constraints, they have a near-optimal performance with a small difference gap when compared to the Oracle baseline. Note that the Myopic approach may fail in some cases(shown in Mate et al. [2020]), it performs worse than the Random approach.

**No penalty for the violation of the fairness constraint:** We also investigate the intervention benefit ratio defined as $\frac{\bar{R}_{\text{method}} - \bar{R}_{\text{No intervention}}}{\bar{R}_{\text{Oracle}} - \bar{R}_{\text{No intervention}}} \times 100\%$, where $\bar{R}_{\text{No intervention}}$ denotes the average reward without any intervention involved. Here, we do not employ penalties when the fairness constraint is not satisfied, as we want to evaluate the benefit provided by interventions with our fair policy and policies of other approaches. We provide the intervention benefit ratio for different values of $L$ for all approaches in Figure 7. Again, the left graph shows the result of FaWT approach, the middle graph is the result of FaWT-U approach, and the right graph shows the result of FaWT-Q method. Our proposed approaches can achieve a better intervention benefit ratio compared with the baseline when $L$ is 30 and above. However, for L = 15, where there is a strict fairness constraint (i.e., $\frac{k \times L}{(\eta-1) \times N}$ is close to 1), it has a significant impact on solution quality. The performances of all our approaches improve when the fairness constraint's strength decreases ($L$ increases). Overall, our proposed methods can handle various levels of fairness constraint strength without sacrificing significantly on solution quality.

We also provide the additional experiment result that studies the influence of intervention level and fairness constraint's strength in the Appendix.

# 6 CONCLUSION

In this paper, we initiate the study of fairness constraints in Restless Multi-Arm Bandit problems. We define a fairness metric that encapsulates and generalizes existing fairness definitions employed for Multi-Arm Bandit problems. Contrary to expectations, we are able to provide minor modifications to the existing algorithm for RMAB problems in order to handle fairness. We provide theoretical results on how our methods provide the best way to handle fairness without sacrificing solution quality. This is demonstrated empirically as well on benchmark problems from the literature.

## Acknowledgements

This research/project is supported by the National Research Foundation, Singapore under its AI Singapore Programme (AISG Award No: AISG2-RP-2020-017).

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
