# OpenReview forum: "Efficient Resource Allocation with Fairness Constraints in Restless Multi-Armed Bandits"
_auai.org/UAI/2022/Conference — UAI 2022 Poster_

### Official Review · Reviewer_vzen · 2022-04-04

**Q2(1) Originality/Novelty:** 3
**Q2(2) Significance/Impact:** 3
**Q2(3) Correctness/Technical Quality:** 3
**Q2(6) Clarity Of Writing:** 3
**Q6 Overall Score:** 7
**Q8 Confidence In Your Score:** 4

**Q1 Summary And Contributions:**

The manuscript proposes new methods for fair policy optimization in the restless multi-armed bandit (RMAB) setting. The authors derive optimality guarantees for several algorithms and demonstrate empirical performance in a range of experiments.

**Q2 Assessment Of The Paper:**

More detailed information regarding each of these aspects is given below:

**Q2(4) Quality Of Experiments (Optional):**

3: Good: The experimental evaluation is adequate, and the results convincingly support the main claims.

**Q2(5) Reproducibility:**

3: Good: Key resources (e.g., proofs, code, data) are available and key details (e.g., proofs, experimental setup) are sufficiently well-described for competent researchers to confidently reproduce the main results.

**Q3 Main Strengths:**

The problem setting is novel and the proposed solutions are convincing. The writing is strong, the technical contributions appear sound, and the experimental results illustrate the performance and limits of the proposed algorithms.

**Q4 Main Weakness:**

If I understand correctly, all three algorithms described here assume that arms are functionally independent. Yet the examples cited at the beginning of the manuscript (e.g., healthcare applications) typically violate this assumption. For example, treatment options may interact or overlap in ways that allow results from one “arm” to inform another.

**Q5 Detailed Comments To The Authors:**

The manuscript is strong, combining novel theoretical and empirical results on an interesting problem that I believe will be of interest to researchers in this area. I therefore recommend acceptance.

My only substantive critique is that the text makes no mention of the possibility that arms may be dependent, and that exploiting such dependencies could lead to better results (e.g., faster convergence, lower cumulative regret, etc.) This has been the focus of the growing literature on causal reinforcement learning, which began with the MAB setting (Lattimore et al., 2016; Lee & Bareinboim, 2018). Causal approaches to fair policy optimization have also been ascendant in recent years (Zhang & Bareinboim, 2018; Nabi et al., 2019; Creager et al., 2020; Huang et al., 2021). While I appreciate that such considerations may fall outside the scope of this manuscript, which is effectively introducing a new method for fair RMABs, some acknowledgment of the problem would be welcome, as it could pose an interesting direction for future research in RMABs.

Minor notes:
-On p. 4, “monotonous” should be “monotonic”
-In Figs. 6-7, the colors for Random and Oracle are much too similar. Please distinguish these more clearly.


**Q7 Justification For Your Score:**

As noted above, the manuscript is strong. Though I am no expert in this field, I was able to follow the discussion and could envision applications to various domains. While I have some reservations about the issue of dependencies between arms, I believe these could be easily addressed with a few sentences at the beginning and/or end of the manuscript.

**Q9 Complying With Reviewing Instructions:**

1: Yes.

---

### Official Review · Reviewer_GYp4 · 2022-04-05

**Q2(1) Originality/Novelty:** 3
**Q2(2) Significance/Impact:** 3
**Q2(3) Correctness/Technical Quality:** 3
**Q2(6) Clarity Of Writing:** 3
**Q6 Overall Score:** 6
**Q8 Confidence In Your Score:** 3

**Q1 Summary And Contributions:**

This paper focused on the Restless Multi-armed bandit problem (RMAB) with fairness constraints. In this work, the author modified the whittle index algorithm with respect to the fairness constraints and provided a theoretical guarantee for optimality. In addition, the experiment results also support the efficiency of this novel algorithm.

**Q2 Assessment Of The Paper:**

More detailed information regarding each of these aspects is given below:

**Q2(4) Quality Of Experiments (Optional):**

3: Good: The experimental evaluation is adequate, and the results convincingly support the main claims.

**Q2(5) Reproducibility:**

3: Good: Key resources (e.g., proofs, code, data) are available and key details (e.g., proofs, experimental setup) are sufficiently well-described for competent researchers to confidently reproduce the main results.

**Q3 Main Strengths:**

1. This paper first considers the fairness constraint within the RMAB problem and proposes a novel algorithm to find the optimal policy.
2. This work considers both finite and infinite horizon cases with a theoretical guarantee. In addition, Experiment results also show that the FAWT algorithm outperforms other baseline algorithms.

**Q4 Main Weakness:**

1. The requirement for Theorem 1 and Theorem 5 seems too strict. The constraint may be satisfied for the simple simulation instance in the experiment. However, it is not clear whether this assumption still holds for some difficult simulation instances or some real-world applications. If not, the results of Theorem 1 and Theorem 5 cannot provide any guarantee for the performance. It is better if the author can give more explanation for those requirements.
2. For the FAWT algorithm (Line 3), the arm $i$ will add to the action set if it is not satisfied at the end of length L. When the FAWT algorithm starts to determine the arm at the $L$-step, the fairness constraint is always satisfied since the number of the previous step is only $L-1<L$, and the number of arms $i$ that is not satisfied at the end of length L may be larger than $k$. In this case, the algorithm will fail whatever the action is chosen, and it is not clear how it works during these situations.

**Q5 Detailed Comments To The Authors:**

For the previous weakness (2), it would be better if the author could give more detail about the algorithm's performance and show how to avoid this situation.

**Q7 Justification For Your Score:**

The motivation for the fairness constraint within the Restless Multi-armed bandit problem is novel and important. In addition, both the theoretical guarantee and experiment result support the efficiency of the FAWT algorithm. However, the main concern is that Theorems' requirements are too restricted to be used in real-world applications.

**Q9 Complying With Reviewing Instructions:**

1: Yes.

---

### Official Review · Reviewer_CiTc · 2022-04-16

**Q2(1) Originality/Novelty:** 2
**Q2(2) Significance/Impact:** 2
**Q2(3) Correctness/Technical Quality:** 3
**Q2(6) Clarity Of Writing:** 3
**Q6 Overall Score:** 6
**Q8 Confidence In Your Score:** 3

**Q1 Summary And Contributions:**

The paper considers solution for RMAB with constraints that the number of arms to be played within a recent fixed epochs. While being "fair to different arms", the proposed algorithm, which utilizes Whittle Index Values and modifies existing algorithm slightly, work desirably under the knowledge of transition. The optimality of the algorithms both in finite and infinite settings are shown. Uncertainty in transition matrix is handled w/ Q-learning and/or Thompson-sampling approach.

**Q2 Assessment Of The Paper:**

More detailed information regarding each of these aspects is given below:

**Q2(4) Quality Of Experiments (Optional):**

3: Good: The experimental evaluation is adequate, and the results convincingly support the main claims.

**Q2(5) Reproducibility:**

4: Excellent: Key resources (e.g., proofs, code, data) are available and key details (e.g., proof sketches, experimental setup) are comprehensively described for competent researchers to confidently and easily reproduce the main results.

**Q3 Main Strengths:**

- The proof for the optimality of the fairness constrained algorithms (finite or infinite).


**Q4 Main Weakness:**

I will cover mostly on Q5.

**Q5 Detailed Comments To The Authors:**

- I am not sure why it is called "fair". Surely "fairness" can be defined differently for different tasks and different problems. In the RMAB setting, the authors defined that it is fair to visit arm that are not visited long time. This is very similar to job scheduling based on "Priority Scheduling" combined with "Longest Job First" (with an exception that we have k actions to be activated at one).

- The introduction (the first page right column last paragraph) tries to motivate RMAB + Fairness constraint by applications for public health. Yet, I really couldn't understand the two long sentences. What do you mean by "never talking to public health workers"? and moving to "bad states"? While it is alluded that arms are "patients, pregnant mothers, etc" yet there is no concrete idea/example to grasp the setting.  (During the reading of the paper, I was keep thinking about an arm as treatment. Maybe this is the reason why I am keep confused about the fairness. Still, the abstract/introduction should be improved.)

- Problem Description is a bit frustrating. Are and States are somewhat coupled but not presented clearly what do we mean by "arms evolving". Arm is a patient but Arm is also the state of the patient. This also makes me confused when omega is belief state and also expected reward (in Eq 4).

- How should you evaluate the novelty of 4.3 and 4.4? Further, the Sec 4.3 and 4.4 both are about the uncertainty/unknown of transition matrix where 4.3 is about modeling the transition and 4.4 is about model-free approach. Section titles can be appropriately changed.

- I wonder whether there is any ways to incorporate fairness constraints naturally as reward (like upper confidence bounds like idea there is increased reward for unvisited individual, it may be hard to incorporate "community"-level constraint into the reward structure)

**Q7 Justification For Your Score:**

Even though the authors proposed intuitively-appealing (and can be quite obvious) algorithms, it would be non-trivial to show the optimality under the newly proposed fairness constraints. However, the novelty and its practical impact seems limited. I hope I am misunderstood and wish the authors can rebut on these points.



**Q9 Complying With Reviewing Instructions:**

1: Yes.

---

### Decision · Program_Chairs · 2022-05-15

**Decision:**

Accept (Poster)

**Comment:**

Meta Review: The reviewers are in agreement with accepting the paper. There are some concerns around the exposition, especially around using the fairness term: we strongly encourage the authors to consider this in the final version.